# Comment on 'Naked mole-rat mortality rates defy Gompertzian laws by not increasing with age'

**Philip Dammann[1,2†]\*, André Scherag[3†], Nikolay Zak[4], Karol Szafranski[5], Susanne Holtze[6], Sabine Begall[1], Hynek Burda[7], Hans A Kestler[5,8], Thomas Hildebrandt[6], Matthias Platzer[5]**

[1]Department of General Zoology, Faculty of Biology, University of Duisburg-Essen, Essen, Germany; [2]University Hospital, University of Duisburg-Essen, Essen, Germany; [3]Institute of Medical Statistics, Computer and Data Sciences (IMSID), Jena University Hospital, Jena, Germany; [4]Moscow Society of Naturalists, Moscow, Russia; [5]Leibniz Institute on Aging – Fritz Lipmann Institute, Jena, Germany; [6]Department of Reproduction Management, Leibniz Institute for Zoo and Wildlife Research, Berlin, Germany; [7]Department Game Management and Wildlife Biology, Faculty of Forestry and Wood Sciences, Czech University of Life Sciences, Prague, Czech Republic; [8]Institute of Medical Systems Biology, Ulm University, Ulm, Germany

**Abstract** Ruby et al. recently analyzed historical lifespan data on more than 3200 naked mole-rats, collected over a total observation period of about 38 years (Ruby et al., 2018). They report that mortality hazards do not seem to increase across the full range of their so-far-observed lifespan, and conclude that this defiance of Gompertz's law 'uniquely identifies the naked mole-rat as a non-aging mammal'. Here, we explain why we believe this conclusion is premature.
DOI: https://doi.org/10.7554/eLife.45415.001

**\*For correspondence:**
philip.dammann@uk-essen.de

[†]These authors contributed equally to this work

**Competing interests:** The authors declare that no competing interests exist.

## Introduction

The historical data set analyzed by *Ruby et al. (2018)* is strongly skewed toward animals born in 2008 or later (*Figure 1*); thus, the observation time for most (87%) of the animals is no longer than 8 years. The reason for this skew – a massive expansion of the colony after 2007 – has been diligently explained by Ruby et al. However, one fundamental consequence of this skew is that the true informative value of the dataset regarding the key claim – that naked mole-rats do not age across the full range of their potential lifespan – is much smaller than that intuitively perceived by the large number of data points (>3,000) in their data set.

Eight years correspond to approximately 25% of the maximum lifespan of the species as recorded to date (>33 years). In these young age cohorts, demographic aging (increasing mortality rates) is not expected to occur: as Ruby et al. correctly point out, Gompertz mortality acceleration typically has its onset in the second half of the potential lifespan of a mammalian species (see also *Jones et al., 2014*). Therefore, looking especially at older age cohorts is crucial for addressing the question of whether a species exhibits Gompertzian aging or not. If naked mole-rats were typical mammals, one would expect mortality acceleration to become apparent no earlier than at approximately 17 years (~50% of their maximum lifespan as determined to date), or even later if the mole-rats could indeed live considerably longer than has been recorded to date (as suggested by Ruby et al.). However, only 23 individuals (i.e. fewer than 1%) were observed for 18 years or longer in the

most stringent analysis by Ruby et al. (their Figure 1), and all of them were subsequently right-censored. Consequently, Ruby et al. limit their conclusions to the first 18 years in some passages ('The mortality hazard of naked mole-rats failed to increase for at least 18 years'), or the first 12 years in other passages ('Our analyses [...] confidently revealed a lack of demographic aging up to at least 4400 days of life (~12 years).'). Although we agree with this approach, we also believe that other statements (such as 'this mouse-sized rodent exhibited no increase in mortality hazard, that is, no Gompertzian aging, across its full, as-yet-observed, multi-decade life-span') are exaggerated at this point, because 'full, as-yet-observed' means more than 33 years.

Because of the small number of data points for mole-rats older than 18 years, Ruby et al. calculated one single hazard estimate for the entire final age group (>18 years). The fact that this hazard estimate was not higher than the hazard estimates for the preceding age groups is surely a strong argument for the interpretation put forward by Ruby et al. However, the decisive question of whether mortality hazards increase toward the end of that age group or not must, by definition, remain unanswered by this approach. Considering also the decrease in survival at approximately 29 years in Figure 2A of Ruby et al., and the unavoidably high statistical uncertainty toward the end of that final age group, we believe that the chance of not-yet-detected aging in later years – which has been acknowledged in principle by Ruby et al. – is too high to allow us to state at this stage that naked mole-rats are 'non-aging mammals'.

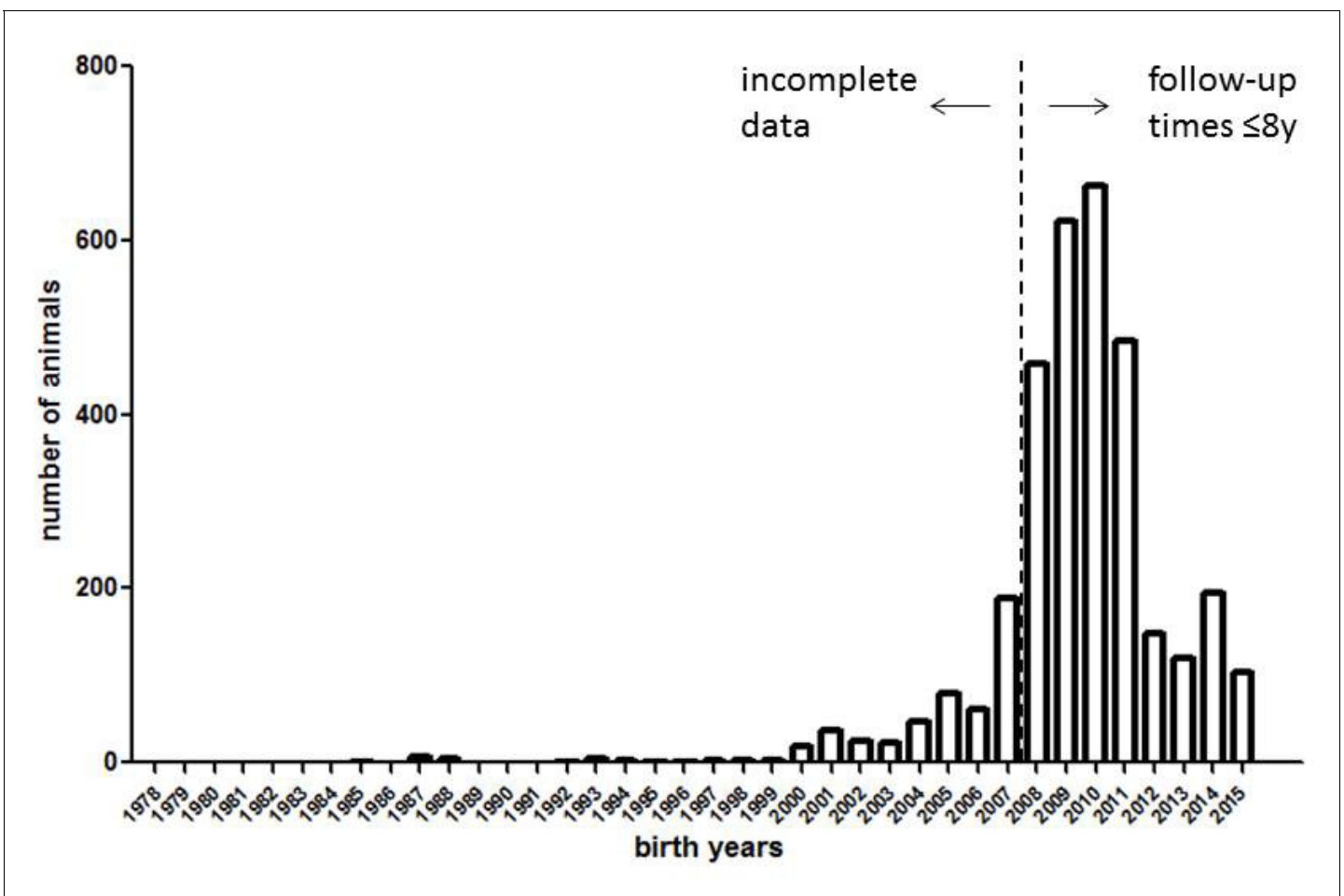

**Figure 1.** Histogram of birth years in the naked mole-rat dataset (3299 data points) underlying Figure 1 of *Ruby et al. (2018)*.
DOI: https://doi.org/10.7554/eLife.45415.002

## Results and discussion

### Bias due to missing death records

Working with the raw data of Ruby et al. revealed that for the observation period of July 2, 1978, through January 30, 2008, that is for nearly 30 years (~76% of the entire observation period), no deaths were reported. Because it is extremely unlikely that no animal died over such a long period of time, we contacted Ruby et al. and expressed our concern that the data set could be biased due to missing death records before 2008. They responded that indeed only animals that had survived after January 30, 2008, and those born after that date were included in the study. They further argued that additional (unpublished) left-censorship-analysis performed by them had satisfactorily controlled for the missing death records between 1978 and 2008, and that their conclusion about non-increasing hazard had not been modified by this alternative analytical approach (Ruby et al., personal communication).

We have not seen this analysis yet, but we doubt that such an approach could truly rescue the situation. Even methods for addressing left censoring require information on the number of missing animals. For the reader, it is hard to evaluate whether this information is available or not, and how much information can still be derived from it. In their supplemental data, Ruby et al. list 250 animals which could not be included in the study due to insufficient resolution (amongst them 37 animals labelled 'death date unknown'), but the reader cannot determine whether these animals represent all missing animals that died between 1978 and January 2008, which age categories they might belong to and so on.

In our view, these aspects need to be addressed because selective reporting of death data can clearly bias all relevant hazard and Kaplan–Meier estimates (underestimation of hazard rates, overestimation of median survival). Since death events at ages up to 29 years may be missing, while no deaths occurring later than 6529 days (~18 years) are recorded in the most stringent analysis by Ruby et al. (their Figure 1A), the bias could be relevant, especially with regard to hazard estimates for older age groups.

We tried to exemplify the potential bias by modeling own lifespan data for related *Fukomys* mole-rats (*Figure 2A*) and using this dataset to 'simulate' the loss of death data over the first 75% (1984–2010) of the total observation period (1984–2018). The result of our simulation is depicted in *Figure 2B*. Not surprisingly, the slope of the survival curve decreases, median survival increases by more than 30%, and the estimated constant mortality hazard per day decreases by more than 25%, from $2.6 \times 10^{-4}$ per day (Figure 3A) to $1.9 \times 10^{-4}$ per day (Figure 3B). Although this is just a single example and both data sets may quantitatively react differently to the introduced bias, these results indicate that the 'consistent estimate for mortality hazard ($8 \times 10^{-5}$ per day)' for the naked mole-rat (*Ruby et al., 2018*) is probably underestimated. Hazards may remain constant with age, but even this possibility remains, in our view, unanswered to date.

### Homologies to other social mole-rats

Of note, we believe that Figure 3 of Ruby et al. contains an important finding because it shows for the first time that, under laboratory conditions, naked mole-rat breeders appear to live longer than helpers. Until recently, only data from the wild suggested this difference in life expectancy for naked mole-rats (*Braude, 1991*; *Buffenstein, 2008*). If corroborated by a stricter, unbiased analysis, this pattern would show a striking similarity to the divergent survival trajectories of breeders and non-breeders in several *Fukomys* species (*Dammann and Burda, 2006*; *Dammann et al., 2011*; *Schmidt et al., 2013*) and thus would support the hypothesis that many if not all social mole-rat species may have evolved life-prolonging mechanisms associated with sexual activity, breeding, or both, despite the classic trade-off between reproduction and somatic maintenance.

### Conclusions

In summary, we argue that the lifespan data of *Ruby et al. (2018)* have to be used and interpreted with caution because they are skewed and biased. Even if naked mole-rats defy the Gompertz–Makeham timeline over large portions of their documented lifespan, they may nonetheless age afterwards (see, for example, *Edrey et al., 2011* for signs of age-associated pathologies in naked mole-rats older than 28 years). In fact, the scientific literature regarding naked mole-rats describes a

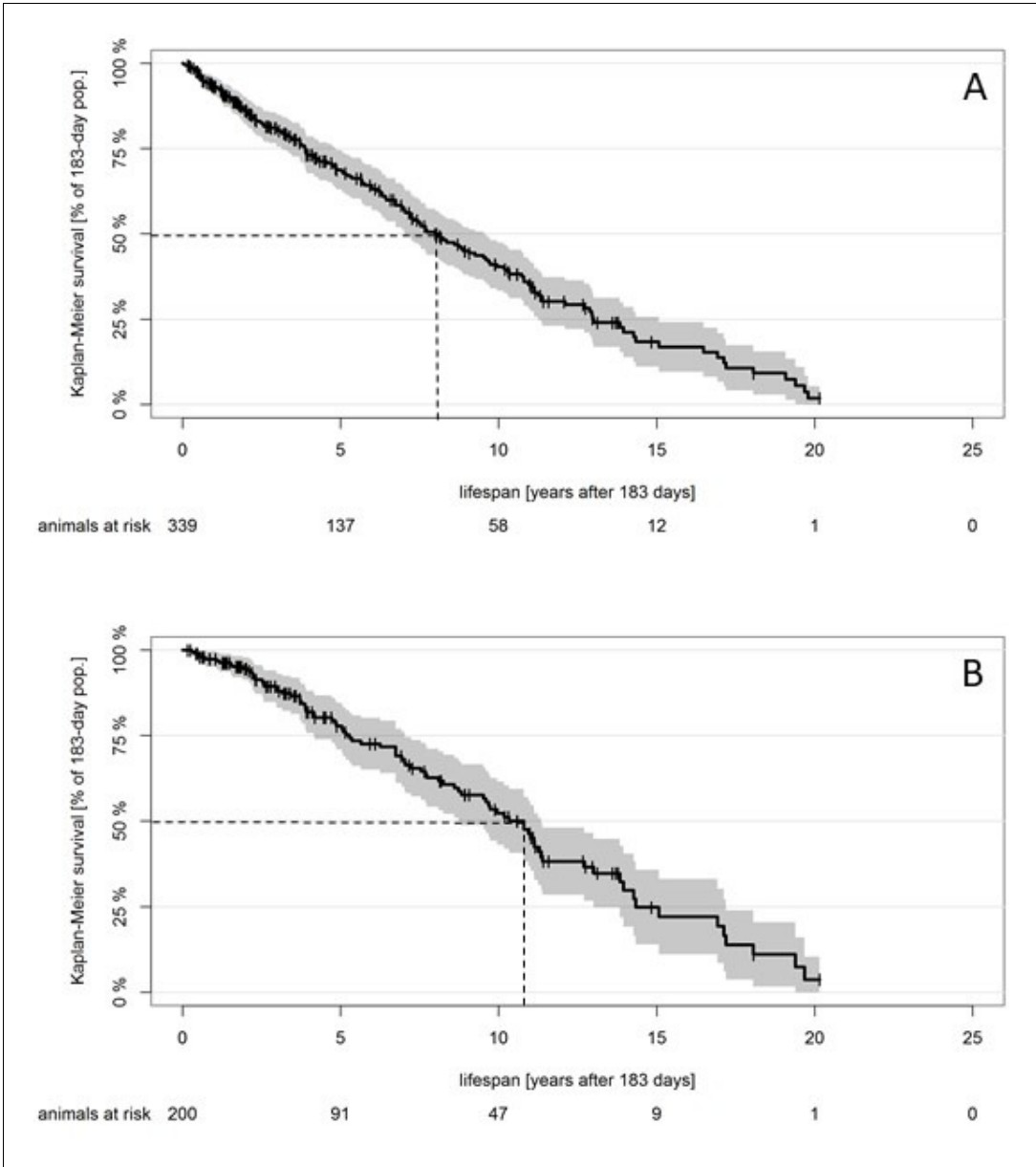

**Figure 2.** Survival curve for small Zambian mole-rats. Kaplan–Meier survival curve for small Zambian mole-rats (*Fukomys anselli* and *Fukomys anselli x kafuensis*) that reach the same age as that used as a starting point in *Ruby et al. (2018)*; 95% confidence intervals for the Kaplan–Maier curve and animals at risk are included, as suggested by *Pocock et al. (2002)*. Dotted lines represent median survival after onset of the study at 0.5 years. (**A**) Original data from 339 animals; median survival after 0.5 years, 7.99 years (95% CI, 7.04–9.60 years). (**B**) Biased data presentation: animals that died before 2010 have been (artificially) deleted from the dataset. Median survival after 0.5 years, 10.80 years (95% CI, 8.82–11.39 years).

DOI: https://doi.org/10.7554/eLife.45415.003

The following source data and source codes are available for figure 2:

**Source code 1.** This script reads the data from *Figure 2—source data 1* to create the reported Kaplan–Meier estimators (*Figure 2A,B*).

DOI: https://doi.org/10.7554/eLife.45415.004

**Source data 1.** This xlxs-file contains the lifespan data for small Zambian *Fukomys*-mole rats that underlie *Figure 2A,B*.

DOI: https://doi.org/10.7554/eLife.45415.005

number of aging phenotypes for these mammals (see *Edrey et al., 2011*; *Beltrán-Sánchez and Finch, 2018*; *Finch, 2009*; *Heinze et al., 2018* and references therein); these phenotypes should be taken into account in discussions concerning whether naked mole-rats are non-aging mammals.

## Additional information

### Funding

| Funder | Grant reference number | Author |
| --- | --- | --- |
| Deutsche Forschungsgemeinschaft | DA 992/3-1 | Philip Dammann |
| Deutsche Forschungsgemeinschaft | PL 173/8-1 | Matthias Platzer |

The funders had no role in study design, data collection and interpretation, or the decision to submit the work for publication.

### Author contributions

Philip Dammann, Conceptualization, Data curation, Supervision, Writing—original draft, Writing—review and editing; André Scherag, Conceptualization, Formal analysis, Writing—original draft; Nikolay Zak, Conceptualization, Investigation; Karol Szafranski, Conceptualization, Writing—original draft; Susanne Holtze, Thomas Hildebrandt, Conceptualization, Resources; Sabine Begall, Hynek Burda, Data curation, Validation, Writing—review and editing; Hans A Kestler, Conceptualization, Formal analysis, Validation; Matthias Platzer, Conceptualization, Supervision, Writing—original draft

### Author ORCIDs

Philip Dammann (iD) https://orcid.org/0000-0002-0624-3965
Nikolay Zak (iD) http://orcid.org/0000-0001-5355-7249
Karol Szafranski (iD) http://orcid.org/0000-0001-6391-1766
Susanne Holtze (iD) http://orcid.org/0000-0002-4654-1916
Sabine Begall (iD) http://orcid.org/0000-0001-9907-6387
Hans A Kestler (iD) http://orcid.org/0000-0002-4759-5254
Thomas Hildebrandt (iD) http://orcid.org/0000-0001-8685-4733
Matthias Platzer (iD) http://orcid.org/0000-0003-0596-8582

### Decision letter and Author response

Decision letter https://doi.org/10.7554/eLife.45415.008
Author response https://doi.org/10.7554/eLife.45415.009

## Additional files

### Data availability

All data generated or analysed during this study are included in the manuscript and supporting files. Source data files have been provided for Figure 2.

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
