## [Decision Letter]

Thank you for submitting your article 'Comment on 'Naked Mole-Rat mortality rates defy Gompertzian laws by not increasing with age' to *eLife* for consideration as Scientific Correspondence. Your article has been reviewed by two peer reviewers, and the evaluation has been overseen by a Reviewing Editor (Michael Rose), a Senior Editor (Patricia Wittkopp) and the *eLife* Features Editor (Peter Rodgers). The following individuals involved in review of your submission have agreed to reveal their identity: Caleb Finch (Reviewer #1); Laurence Mueller (Reviewer #2).

The reviewers supported publication of your article subject to the following points being addressed in a revised version.

Essential revisions:

While we think using the data on Zambian mole-rats is an interesting way of illustrating the effects of possible bias, it has limitation. Essentially, it is a single example of the effects of left censoring data. A more useful analysis would exploit simulated death records and vary the timing, number of censored records, etc. For a comment note, this type of simulation is not needed, but the authors might reflect on the fact that the bias they observed may be affected by specific details of the database used. The bias in the naked mole-rat database can certainly be different.

---

## [Author Response]

Essential revisions:While we think using the data on Zambian mole-rats is an interesting way of illustrating the effects of possible bias, it has limitation. Essentially, it is a single example of the effects of left censoring data. A more useful analysis would exploit simulated death records and vary the timing, number of censored records, etc. For a comment note, this type of simulation is not needed, but the authors might reflect on the fact that the bias they observed may be affected by specific details of the database used. The bias in the naked mole-rat database can certainly be different.

We agree that our simulation using Zambian mole-rats has its limitations, and that the bias in the naked mole-rat data base might be different. We acknowledge this more explicitly than before in our updated text.